# Effect of Immobilization of Phenolic Antioxidant on Thermo-Oxidative Stability and Aging of Poly(1-trimethylsilyl-1-propyne) in View of Membrane Application

**DOI:** 10.3390/polym14030462

**Published:** 2022-01-24

**Authors:** Sergey Shishatskiy, Vladimir Makrushin, Ivan Levin, Petra Merten, Samira Matson, Valeriy Khotimskiy

**Affiliations:** 1Helmholtz-Zentrum Hereon, Institute of Membrane Research, Max-Planck-Str. 1, 21502 Geesthacht, Germany; petra.merten@hereon.de; 2A.V. Topchiev Institute of Petrochemical Synthesis of Russian Academy of Sciences, Leninsky Prospect 29, 119991 Moscow, Russia; makrushin@ips.ac.ru (V.M.); levin@ips.ac.ru (I.L.)

**Keywords:** poly(1-trimethylsilyl-1-propyne), thermal stability, aging, annealing, high free volume polymer, gas permeation

## Abstract

The effect of phenolic antioxidant Irganox 1076 on the structure and gas permeation behavior of poly(1-trimethylsilyl-1-propyne) (PTMSP) was investigated. Isotropic films as well as thin film composite membranes (TFCM) from pure PTMSP and with added antioxidant (0.02 wt%) were prepared. PTMSP with antioxidant has a significantly higher thermal degradation stability in comparison to pure polymer. The thermal annealing of isotropic films of PTMSP with antioxidant was carried out at 140 °C. It revealed the stability of gas permeation properties for a minimum of up to 500 h of total heating time after a modest permeation values decrease in the first 48 h. X-ray diffraction data indicate a decrease in interchain distances during the heat treatment of isotropic films and indicate an increase in the packing density of macromolecules during thermally activated relaxation. Isotropic films and TFCMs from pure PTMSP and with antioxidant stabilizer were tested under conditions of constant O_2_ and N_2_ flow. The physical aging of thick and composite PTMSP membranes point out the necessity of thermal annealing for obtaining PTMSP-based membranes with predictable properties.

## 1. Introduction

Glassy poly(1-trimethylsilyl-1-propyne) (PTMSP) has an extremely high fractional free volume (>0.3), which determines its extremely high permeability to gases and vapors, as well as the highest selectivity of condensable hydrocarbons over permanent gases, which is necessary for the recovery of organic vapors [1,2,3,4,5]. Bulky substituents at alternating double bonds of rigid chains prevent the dense packing of PTMSP macromolecules and form a microporous morphology with a high specific surface area according to Brunauer–Emmett–Teller analysis [6,7].

Due to the nonequilibrium state of PTMSP, physical aging is observed, which is a characteristic property of glassy polymers. The relaxation of macromolecular chains over time leads to a noticeable change in the characteristics of the polymer [8,9,10,11]. As a consequence, aging leads to a decrease in the free volume and changes in the gas transport parameters of the polymer [9,12]. Often, along with a drop in permeability, a significant increase in selectivity is observed. For example, an increase in O_2_/N_2_ and H_2_/N_2_ selectivity along with a decrease in permeability was noted for aged PTMSP films [13,14]. This fact should be attributed to a drop in the diffusion coefficients due to an increase in the density of the polymer with time, which is confirmed by a decrease in the interchain distance in the polymer. An increase in the O_2_/N_2_ selectivity with a general decrease in gas permeability was observed as a result of thermal treatment of the PTMSP film for 15 h at 100 °C [15]. The authors also interpreted the results from the standpoint of relaxation and an increase in the polymer packing density under the action of heating. Moreover, it was shown in [16] that PTMSP, selective in favor of condensable gases such as CO_2_ and H_2_S, exhibits accelerated physical aging at elevated temperatures and acquires selectivity in favor of non-condensable H_2_. The data presented indicate the possibility of increasing the selective characteristics of the polymeric membrane material due to the controlled thermally activated relaxation of glassy polymers with a high free volume.

Furthermore, the investigation of PTMSP aging behavior at high temperatures is very important for searching for opportunities of keeping membrane polymer characteristics stable through the membranes’ lifetime. The effect of elevated temperatures on the enhanced physical aging of PTMSP has already been reported [17]. Thus, the solvation of this problem is necessary in order to ensure sustainable membrane operation, especially at elevated temperatures and at aggressive working conditions.

PTMSP is characterized by high thermal stability due to double bond shielding by bulky substituents of the main chain in the helical conformation [18]. Nevertheless, when heated in air, PTMSP is subject to degradation due to oxidation. The oxidation and destruction of carbochain polymers was described by the theory of radical chain processes of organic substrate oxidation. A possible mechanism for the oxidation of disubstituted polyacetylenes was proposed in the literature [19]. It consists of several stages: (1) formation of biradicals; (2) transformation of biradicals into monoradicals; (3) formation of hydroperoxides by the reaction of polymer radicals with oxygen molecules; and (4) cleavage of a bond in the polymer chain with the formation of C = O and C–OH groups. Nagai and Nakagawa investigated the oxidation of PTMSP in detail and suggested that some unpaired electrons exist on the backbone chain, and they can be transferred to an allyl-type –CH_3_ group [20]. Therefore, the onset of the PTMSP oxidation process in air is observed already at room temperature, however, without a molecular weight decrease. Although some unpaired electrons also seem to exist at the C=C bonds of the backbone chain, oxidation with destruction of the main chain is hardly noticeable at room temperature due to the effective screening of the folded macrochain by bulky substituents and insufficient energy for the oxidation reaction to proceed at the double bond. In the case of PTMSP due to oxidation at elevated temperatures, the decomposition started in the backbone chain.

Disubstituted polyacetylenes with high free volume have been the subject of research for several decades but have not found wide practical application. Their inherent instability and rapid aging prevents them from being used in real membrane separation processes. Thermal annealing at the stage of membrane formation can make it possible to find a way to control and stabilize the properties of membranes based on disubstituted polyacetylenes with a high free volume.

In this paper, we studied the effect of adding phenolic antioxidant stabilizer Irganox 1076 on the structure and gas permeation behavior of PTMSP. The addition of antioxidants makes it possible to slow down significantly the oxidative destruction that polymers undergo under thermal, chemical, and radiation influences [21]. The inhibitory effect of stabilizing antioxidants is to accept alkyl and peroxide radicals. Hindered phenolic compounds are widely used as polymer stabilizers. Irganox 1076 is a highly effective phenolic antioxidant stabilizer. It is used to protect polymers, such as polyolefins, from thermal oxidative degradation. This antioxidant is characterized by compatibility to various polymers, e.g., polyethylene, polystyrene, etc., and good thermal stability.

The effect of thermal annealing on the gas transport properties of isotropic films prepared from PTMSP with the addition of antioxidant Irganox 1076 in comparison with pure PTMSP films was investigated. Moreover, some authors observed that the physical aging of thin films is much more rapid than in thick films [22,23]. Thus, we also explored the effect of Irganox 1076 presence on stability of thin film composite membranes (TFCM) of PTMSP and various physical properties of polymer, including gas transport.

## 2. Materials and Methods

### 2.1. Materials

Monomer 1-trimethylsilyl-1-propyne was obtained by reacting methylacetylene from the methylacetylene-allene fraction with an alkyl magnesium halide, which is followed by treating the reaction mixture with trimethylchlorosilane [24]. Monomer (99.8%) and the solvent cyclohexane (99.8%, Fisher Scientific, Waltham, MA, USA) were distilled three times over calcium hydride in a high-purity argon atmosphere before polymerization. Catalyst NbBr_5_ (99.9%, ABCR, Karlsruhe, Germany), cocatalyst Ph_4_Sn (>98.0%, TCI Ltd., Tokyo, Japan), and antioxidant Irganox 1076 (Octadecyl β—(3,5-di- tert -butyl-4-hydroxy phenyl)-propionate) (Sigma-Aldrich Chemie GmbH, Steinheim, Germany) were used without further purification.

O_2_, N_2_ with purity at least 99.999%, Ar with purity at least 99.9999%, and CO_2_ with purity at least 99.99% were purchased from Linde (Linde GmbH, Pullach, Germany; Linde Gas, Balashiha, Russia).

### 2.2. PTMSP Synthesis and Characterization

#### 2.2.1. Synthesis

The polymerization was carried out in an inert atmosphere of high-purity Ar according to the procedure described elsewhere [25]. PTMSP was synthesized using an NbBr_5_/Ph_4_Sn catalyst in cyclohexane at 80 °C at an initial monomer concentration [Mon]_0_ = 1 mol/L and a monomer/catalyst ratio [Mon]/[Cat] = 50. 

#### 2.2.2. Intrinsic Viscosity

The intrinsic viscosity values of the polymers were measured in CCl_4_ at 25 °C in air, at atmospheric pressure using an Ostwald–Ubbelohde viscometer. For the polymer solution, a weighed portion of 0.01 g per 10 mL was taken and dissolved for 24 h at room temperature. 

#### 2.2.3. ^13^C NMR Spectroscopy

^13^C NMR spectra of solutions of polymers in C_6_D_12_ were recorded on a spectrometer of the Avance series (Bruker BioSpin GmbH, Ettlingen, Germany) in a single-pulse mode with broadband decoupling from protons during a free induction decay signal and a reduced decoupling power during the relaxation delay (duration 3 s), which allows maintaining and accounting for the Overhauser effect. The spectral capture width was 250 ppm. The accumulation time of the free induction decay signal was 12–18 h. Spectra were analyzed, including the decomposition of complex spectral lines, using the ACD Labs program (Advanced Chemistry Development, Inc., USA, version 12.0 for Microsoft Windows). Determination of the quantitative ratio of *cis*- and *trans*-units in polymer samples was performed using ^13^C NMR spectra (Appendix A in Supporting Information), as described in our previous work [25]. The calculation was carried out with respect to the peak intensities with chemical shifts: 152.0–152.8 ppm (*cis*-configuration) and 151.0–151.3 ppm (*trans*-configuration) for a signal corresponding to a carbon atom =C–CH_3_; and 139.6–140.0 ppm (for *cis*-configuration) and 137.7–138.5 ppm (*trans*-configuration) for a signal corresponding to a carbon atom =C–Si.

### 2.3. Preparation and Characterization of Films and Membranes

#### 2.3.1. Preparation of PTMSP Isotropic Films

Isotropic films of PTMSP and PTMSP mixed with Irganox 1076 were prepared by two participating research groups using slightly varying methods.

Method 1. Isotropic films were obtained by casting from a polymer solution (3 wt% of polymer in cyclohexane) onto a leveled surface of stretched cellophane. The films were dried at ambient conditions for 14 days and then in a vacuum at room temperature for 48 h. The thickness of the obtained films was 30–50 µm. 

Isotropic films from PTMSP with Irganox 1076 were obtained by a similar procedure from polymer solutions in cyclohexane, to which 0.02 wt% of antioxidant in respect to polymer was added. 

Method 2. Another set of films was prepared by a similar procedure but with a Teflon^®^-coated surface was used for polymer solution casting. In this case, 0.15 wt% polymer solution in cyclohexane was prepared, filtered, and poured into the evaporation container with a polished Teflon^®^ bottom; then, the container was closed with a glass lead having connections for nitrogen inlet and outlet. Polymer solution was dried under N_2_ flow of 10 mL/min for 48 h, and when no presence of solvent was detectable, the polymer film was mechanically removed from the Teflon^®^ surface and placed into desiccator with silica gel and activated carbon functioning as adsorbents for water and organic vapors. All films were prepared from the same amount of solution and had thickness of 23–25 µm. 

#### 2.3.2. TFCM Formation

TFCMs with a selective layer of PTMSP were prepared by the modified deep-coating method described elsewhere [26]. Casting solution with polymer concentration of 0.5 wt% in cyclohexane was stirred for 24 h to ensure full polymer dissolution, filtered, and poured into the coating bath. A polyacrylonintrile (PAN) ultrafiltration membrane on polyester nonwoven was used as a support for selective layer. Polymer solution was bought into contact with the PAN membrane, removed from the PAN surface for 2 mm to form a meniscus, and the PAN membrane was drawn through the meniscus of polymer solution to form a thin solution layer on the porous surface. The obtained membrane was dried at ambient conditions for 1 h and placed into a desiccator as described above. No thermal or other treatments were done to membranes prior to gas transport experiments.

#### 2.3.3. Thermal Treatment of PTMSP Isotropic Films

Thermal treatment of polymer films was carried out in air in a BINDER FD 53 drying oven (Binder GmbH, Tuttlingen, Germany) equipped with an electronic controller allowing for temperature stability with an accuracy of 1 °C. The oven was heated electrically, and the uniformity of the temperature profile within the heated chamber was ensured by air circulation within the oven by means of an air blower. The inner chamber, the pre-heating chamber, and the inner side of the doors are made of stainless steel (material # 1.4301). The polymer films in glass Petri dishes were placed into the oven, after which the chamber was heated. The set temperature was reached within 15–20 min. Upon the predetermined heating time of 24 h, the films were cooled in an oven for 3 h and then tested for the permeability of O_2_, N_2_, and CO_2_. 

#### 2.3.4. Film Density Determination

A simple geometric approach for density determination was chosen. Prepared films were cut using ring knives to samples with diameters necessary for gas transport properties determination. The film thickness was determined using a Fischer Deltascope FMP10 (Helmut Fischer GmbH, Sindelfingen, Germany) at as many points along the surface of the sample as possible. Samples weight determined on analytical balance (Sartorius analytic 200S, Sartorius Weighing Technology GmbH, Göttingen, Germany) as well as sample volume obtained from diameter and thickness were used for calculation of density. 

Density determination by the buoyancy method usually gives more accurate results, but it was not used for the samples under investigation due to the very high static charge of the samples, which was not possible to dissipate by immersing into the 3M™ Fluorinert™ FC-770 fluid (3M Deutschland GmbH, Neuss, Germany). Additionally, some unexpected interaction between FC-770 and PTMSP films was observed, resulting in a significant change of film shape.

#### 2.3.5. Thermogravimetric Analysis

Thermogravimetric analysis (TGA) was carried out using a TG 209 F1 Iris (Netzsch, Selb, Germany). The experimental conditions were as follows: temperature range 20–600 °C, heating rate 10 K/min, nitrogen atmosphere.

#### 2.3.6. Wide-Angle X-ray Diffraction (WAXD)

Diffractograms of polymer films were obtained on a Rigaku Rotaflex RU-200 X-ray diffractometer with a rotating copper anode (characteristic radiation wavelength 0.1542 nm). Flat films were fixed in 6 layers on an aluminum frame. The survey was conducted in the “on-passage” mode in the range of diffraction angles of 2.5–50 degrees at 2θ according to the Bragg–Brentano scheme, which is characterized by the constancy of the magnitude of the irradiated sample volume regardless of the diffraction angle. Then, the obtained diffractograms were processed using the Fityk program: after subtracting the background line, they were represented as the sum of several Gaussian peaks. The position of these peaks according to the Wulff–Bragg formula was recalculated into the interchain distance. The specific intensities of the maxima were calculated from the peak areas, and the sizes of their coherent scattering regions were estimated using the Selyakov–Scherrer formula.

#### 2.3.7. Gas Transport Properties Determination

##### Determination of Gas Permeability for PTMSP-Based Isotropic Films Prepared by Method 1

The permeability coefficients for individual gases were determined at 20 °C on a unit operating according to the “constant volume/variable pressure” principle. The work is based on the manometric method for measuring the flow rate of gas that has passed through the membrane. The flow rate of the gas passing through the membrane is determined by the time it flows into the calibrated, evacuated volume. The membrane permeability coefficients were calculated using the equation:(1)$$P=\frac{V\times \Delta p\times l}{t\times A\times p\times 760\times 76}\times 10^{10}$$
where *P* is the permeability coefficient (Barrer), *V* is the calibrated permeate volume (1175 cm^3^); *A* is the membrane area (24.18 cm^2^); *l* is the membrane thickness (cm); *p* (atm) is the pressure drop across the membrane (up to 10 atm); and *t* is the time of gas inflow into the permeate volume *V* from pressure *p*_1_ to pressure *p*_2_ (the difference Δ*p* (torr) = *p*_2_ − *p*_1_ can be selected from the series 1, 2, 5, 10, 20 torr), which determine the moments when the stopwatch is turned on and off.

##### Determination of Gas Permeability for Isotropic Films Prepared by Method 2 and Gas Permeances of TFCMs

Gas transport properties of 25 µm thick films and TFCMs for O_2_ and N_2_ were carried out on the experimental facility equipped with 47 mm inner diameter measurement cells. The facility operates according to the “constant pressure/variable volume” measurement principle, and the permeate flow rate was determined with a high-accuracy positive displacement flow meter (Bios DryCal Definer 220, MesaLabs Inc., Lakewood, CO, USA). Experiments were carried out at room temperature for 2 weeks. During the 1st day of the experiment, N_2_ and O_2_ were flown through the sample in change in order to obtain 2–3 experimental points. During the rest of the 1st week, N_2_ was constantly flowing through the sample under investigation, and once per day, the permeate flow was determined and N_2_ was exchanged to O_2_. After 1 h, when N_2_ on the permeate side of the sample was exchanged to O_2_, the permeate flow rate was determined, and the gas was changed to N_2_ again. During the 2nd week of the experiment, gases were exchanged, and O_2_ was constantly flowing through the sample. Experiments were carried out at 2 bar pressure gradient across the thick film and 1 bar gradient across the TFCM.

The permeability coefficient was calculated according to Equation (2):(2)$$P=\frac{V\times l}{A\times t\times \left(p_{feed}-p_{permeate}\right)}\times 10^{10}$$
where *P* is the permeability coefficient (Barrer), and *V* is the volume of gas (cm^3^ (STP)) permeating through the sample with thickness *l* (cm^2^) in a unit of time *t* (s) at the active sample area *A* (cm^2^) under driving force of pressure gradient *p_feed_* − *p_permeate_* (cm Hg) across the sample.

TFCM permeance was measured at the same facility and calculated according to Equation (3):(3)$$L=\frac{V}{A\times t\times \left(p_{feed}-P_{permeate}\right)}$$
where *L* is the membrane permeance (m^3^(STP) m^−2^ h^−1^ bar^−1^), and *V* is the volume of gas (m^3^ (STP)) permeating through the sample in unit of time *t* (h) at active sample area *A* (m^2^) under driving force of pressure gradient *p_feed_* − *p_permeate_* (bar) across the sample.

## 3. Results and Discussion

### 3.1. Gas Transport Properties of Isotropic Films

For our experiments, we used PTMSP highly enriched with *cis*-structures, since the stability of *cis*-enriched PTMSP is higher than that of polymer with a mixed geometric structure [18]. The PTMSP used in this work had the intrinsic viscosity [η] = 3.6 dL/g, and ratio of *cis*-/*trans*-units was 80/20. Pure gas permeation properties obtained from experiments with pure PTMSP and PTMSP with added Irganox 1076 (PTMSP-S) are reported in Table 1. One can observe that the introduction of a stabilizer in the amount of 0.02 wt% leads to a decrease in the permeability coefficients in comparison to pure PTMSP film. The decrease in permeability, apparently, is a consequence of a more tortuous pass for the penetrant molecule through the polymer matrix upon the introduction of stabilizer molecules. It should be noted that the values of the permeability coefficients obtained independently in TIPS and Hereon are in good agreement despite differences in film preparation methods and measurement techniques.

### 3.2. Thermal Stability and Aging of Annealed PTMSP Isotropic Films

TGA investigation of isotropic films of pure PTMSP and PTMSP-S with a stabilizer was conducted in inert atmosphere. No evidence of solvent loss was observed before the samples started to decompose (Figure 1). All samples demonstrate thermal stability up to at least 250 °C, 3% weight loss was at temperature above 340 °C, after which temperature a sudden drop of the weight loss curve was observed. This should be referenced to thermal decomposition due to the breaking of all bonds in the polymer.

Figure 2 presents temperatures at which samples lost 1, 3, 5, and 10% of weight. The major difference in temperature was observed for the 1% weight loss. The stabilizing effect of Irganox 1076 on PTMSP revealed during the TGA experiment shows that indeed, some unpaired electrons exist on the polymer chain, and their presence is causing polymer degradation upon heating in non-oxidizing (inert) atmosphere. In order to be able to anneal the polymer, it is important to ensure the proper location of antioxidant molecules within the polymer matrix in the immediate vicinity to possible locations of unpaired electrons. As it will be discussed later, the introduction of antioxidant molecules in the polymer matrix adds to polymer stability and only slightly reduces gas permeability coefficients.

### 3.3. Gas Transport and Structural Changes in Heat-Treated PTMSP Films

In order to study the effect of sterically hindered stabilizer Irganox 1076 on the permeability of PTMSP films upon heat treatment, we thermally treated a film of PTMSP with added stabilizer by contrast to pure PTMSP film and analyzed changes in the gas permeability. Pure gas permeation experiments carried out for films of pure PTMSP and with stabilizer treated at 100 °C are given in Table 2. It can be observed that the addition of Irganox 1076 significantly increases the thermo-oxidative stability of PTMSP. The gas permeability of pure PTMSP film after heating for 24 h decreases, and after 48 h, it begins to increase, indicating the properties’ deterioration with subsequent polymer film destruction, which occurred after 72 h of heating. The film of stabilized PTMSP shows a slight drop in permeability and retains integrity even after 72 h of heating. In order to accelerate the polymer relaxation, PTMSP film with stabilizer was treated at higher temperature. Figure 3 shows pure gas permeability and the ideal selectivity of PTMSP-S film treated at 140 °C. The permeability for all gases decreased by 20–30% from the initial values within the first 48 h, and after that, the permeability remained stable up to 500 h of total heating time. Simultaneously with a decrease in permeability, an increase in the ideal selectivity was observed, which is associated with thermally activated relaxation and the resulting decrease in the free volume fraction and, most probably, a decrease in the size of the of free volume elements (FVE). For the CO_2_/N_2_ pair, the increase in ideal selectivity values is more noticeable than for the O_2_/N_2_ pair, which is associated with a smaller relative drop in CO_2_ permeability (26%) compared to other gases (O_2_ by 30%; N_2_ by 38%) with bigger molecular sizes.

WAXD analysis was carried out for investigation of the changes in the structure of PTMSP-based films. X-ray diffraction patterns of pure and stabilized PTMSP isotropic films show the main reflex with a half-width of Δ_1/2_^0^ ≈ 3.5–3.8° and additional diffuse maxima (Figure 4). The values of the half-widths of the main reflex indicate a relatively small size of coherent scattering regions, but these are still larger than for truly amorphous polymers, for which the half-width of the reflex is usually 5–8° [27]. As one can see from Figure 4, the inclusion of a stabilizer in the PTMSP did not change the angular position of the main maximum. However, after heating, a shift of the main maximum toward larger angles is observed both for pure PTMSP and for stabilized PTMSP-S. In addition, the interchain distance decreased for both heat-treated pure PTMSP as well as stabilized PTMSP (Table 3). This indicates compaction of the system and growth of polymer density during the process of thermal annealing.

### 3.4. Aging of PTMSP Films under the Conditions of Constant O_2_ and N_2_ Flow 

It is interesting to observe the effect of presumably physical aging on the gas transport properties of polymers when O_2_ and N_2_ constantly flow through polymer samples. Films of ca. 25 µm thickness were exposed to constant flow of N_2_ for one week and O_2_ for another week. Once a day, the gas was changed for O_2_ (1st week) and N_2_ (2nd week), and gas flow through all the samples was determined. As it follows from Figure 5, both pure PTMSP film as well as film with stabilizer show an insignificant but constant decrease in permeability coefficients with time. PTMSP in pure and stabilized form show similar permeability coefficients. The O_2_/N_2_ selectivity for both of the films is practically the same and does not change during continuous N_2_ flow (1st week of experiment). During the 2nd week, when O_2_ was continuously flown through the samples, the O_2_/N_2_ selectivity started to increase. The predetermined duration of experiment did not allow revealing the end of changes in the gas transport properties of the studied samples. A comparison of the results depicted in Figure 5 and Table 1 indicates that the initial property of polymer films prepared on two different surfaces from polymer solutions of different concentrations are comparable. The time taken for solvent evaporation in cases of Methods 1 and 2 is very different, 14 and 2 days, and still, the gas transport properties did not become significantly different.

An insignificant effect of Irganox 1076 introduction on permeability coefficient and O_2_/N_2_ selectivity may be interpreted from the location of antioxidant molecules in the FVEs of the polymer matrix. In order to understand where the antioxidant molecules can be located, molecular modeling of antioxidant molecules was carried out using geometry optimization functions of HyperChem 8.0 (Hypercube Inc, Gainesville, FL, USA) software. After full molecule optimization, the QSAR function of HyperChem was called out, and the molecular weight (530.87 g mol^−1^), van der Waal surface area (709.67 Å^2^), and volume (584.08 Å^3^) were calculated. The surface area and volume were recalculated into molecule diameters using an approximation of spherical molecule shape. The resulting molecular diameters were 13.6 Å for the case of the surface and 10.4 Å for the case of the molecules’ volume. The obtained result was used for consideration in which FVEs molecules of this size can be situated in the polymer.

Presumably, large molecules can be located in the FVEs of PTMSP or in the interchain space of the polymer. The latter case cannot be explored by modeling; the free volume case can be investigated taking into account polymer density, free volume amount, and FVE size distribution.

Densities of isotropic films of pure and stabilized PTMSP have similar values, and error ranges for both samples are overlapping, giving no clear picture of antioxidant influence on sample density (Figure 6). The average density for samples was 0.86 g cm^−3^; this value was used for the concentration determination of FVEs big enough to enclose molecules of Irganox 1076.

According to publications by Yampolskii et al. [28,29,30], PTMSP possesses up to 33% of free volume, the elements of which can be divided into four categories. The only category suitable in size for the accommodation of antioxidant molecules is that described by the τ_4_ term of positron annihilation lifetime spectroscopy (PALS) spectra analysis developed for the description of PALS results for high free volume polymers such as disubstituted polyacetylenes, amorphous Teflon^®^ AF2400, PIM, and additional type polynorbornenes. In case of PTMSP, the τ_4_ elements have a characteristic diameter 13.6 Å and are sufficiently large to accommodate antioxidant. FVEs τ_3_ with a diameter of 7.2 Å are too small to accommodate the large Irganox 1076 without changes in polymer chains packing and are not considered in the current study. The fraction of τ_4_ elements is about 8.3% to the volume of polymer, which gives a concentration of 7.5 × 10^19^ cm^−3^. The amount of antioxidant molecules to be fitted into this amount of vacancies is 6.6 × 10^−2^ g cm^−3^ or, taking into account the average density of the studied samples, 0.864 g cm^−3^ 7.64 wt% of antioxidant dispersed in the polymer. It means that just a few τ_4_ elements of the polymer are occupied by antioxidant molecules at 0.02 wt% concentration, causing a minor effect on gas transport properties, but as it was revealed by heat treatment experiments, effectively protecting the polymer from degradation.

### 3.5. Aging of TFCMs

TFCMs with selective layers of pure PTMSP and PTMSP-S, as in the case of thick isotropic films, were exposed to constant flow of N_2_ during the 1st week of the experiment and O_2_ during the 2nd (at ca. 70 h of experiment). The pressure gradient between the feed and permeate sides of membranes was 1 bar. The results of experiment are shown in Figure 7. TFCMs have a clear decay of permeance for both O_2_ and N_2_. N_2_ permeance for PTMSP-S TFCM decreased by a factor of 10, while the O_2_/N_2_ selectivity increased from 1.3 to 1.9.

TFCM with a selective layer of pure PTMSP shows O_2_ and N_2_ permeances higher than those of PTMSP-S, which is similar to the effect of Irganox 1076 revealed for isotropic films. From the comparison of initial TFCM permeances with permeability coefficients of thick films, the apparent thickness of the selective layer can be estimated as 420 ± 50 nm. 

The accelerated aging of TFCMs, in comparison to isotropic films, is a result of the membrane preparation technique where a thin layer of polymer solution is drawn from the meniscus and subjected to 2 processes occurring simultaneously: (a) solvent evaporation into the environment, and (b) the suction of solvent out of solution into a highly porous PAN membrane serving as a support for the selective layer. Pores of the PAN support are in the range of 5–30 nm with an average pore size of 11 nm [26]. High molecular weight polymer cannot penetrate into pores, while the solution is effectively depleted of solvent under the influence of capillary forces. The two processes combined lead to a fast change of polymer state from being dissolved in diluted solution to solid [31]. The process of solution concentration increase is accompanied by a strong surface temperature decrease. All aforementioned events lead to the formation of a polymer layer with excessive non-equilibrium free volume, which is much bigger than that in the case of isotropic films. The presence of high total free volume can be concluded from the initial O_2_/N_2_ TFCM selectivity, which in the beginning of experiment is as low as 1.29 and only after 6 days of aging became similar to that of isotropic film.

During the 1st week of the experiment, membranes under continuous N_2_ flow changed their O_2_/N_2_ selectivity in a regular and similar way. During the next week, when N_2_ was exchanged to O_2_, no change in the pattern either for permeances nor for selectivities was observed. The final α(O_2_/N_2_) for both membranes was significantly higher than that of isotropic films, reaching a value of 1.92 for PTMSP-S TFCM. The experiments with TFCMs clearly show the necessity of accelerated membrane annealing by thermal treatment for the fast and efficient stabilization of membrane properties before practical use start.

## 4. Conclusions

In this work, the effect of immobilization of the sterically hindered phenolic stabilizer Irganox 1076 in PTMSP on polymer thermal stability as well as gas permeation behavior was investigated. Isotropic films as well as TFCMs from pure PTMSP and with added stabilizer were prepared. Structural, thermal, and gas transport properties were investigated. The introduction of an antioxidant significantly improved the thermal stability of PTMSP. The thermal annealing of isotropic films of PTMSP with an antioxidant stabilizer at 140 °C revealed the stability of gas permeation properties during 500 h of total heating after a modest decrease in permeation values during first 48 h. An increase in the ideal selectivity values was observed, which is associated with a decrease in the fractional free volume and in the size of the elements of free volume. The results of isotropic, not thermally treated, films prepared from both pure PTMSP and polymer with a stabilizer exhibited a gradual aging of polymer after testing by exposure to constant flow of N_2_ and O_2_. TFCM prepared from either pure or stabilized PTMSP exhibits rapid physical aging under conditions of constant gas flow. Thus, the introduction of an antioxidant stabilizer makes it possible to increase significantly the oxidative stability of PTMSP, which allows annealing at high temperatures. The use of thermal annealing of films can help find ways for the incorporation of high free volume polymers into practical membranes with predictable properties.

## Figures and Tables

**Figure 1 polymers-14-00462-f001:**
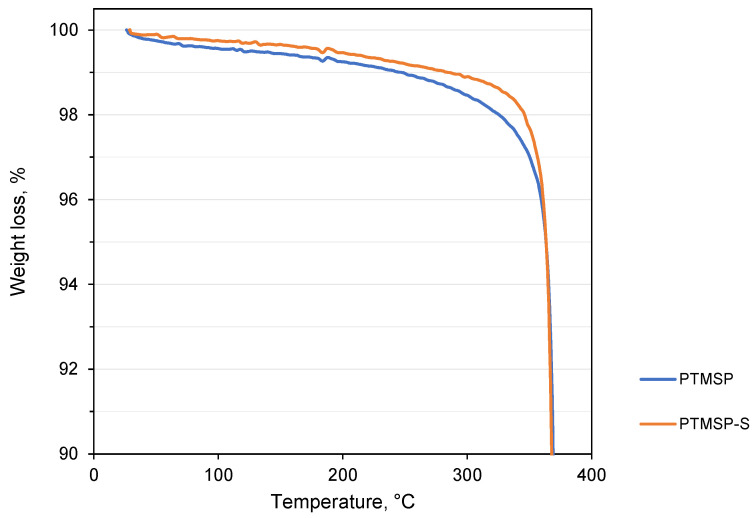
TGA thermographs of materials under investigation in the range of 100–90% of weight loss.

**Figure 2 polymers-14-00462-f002:**
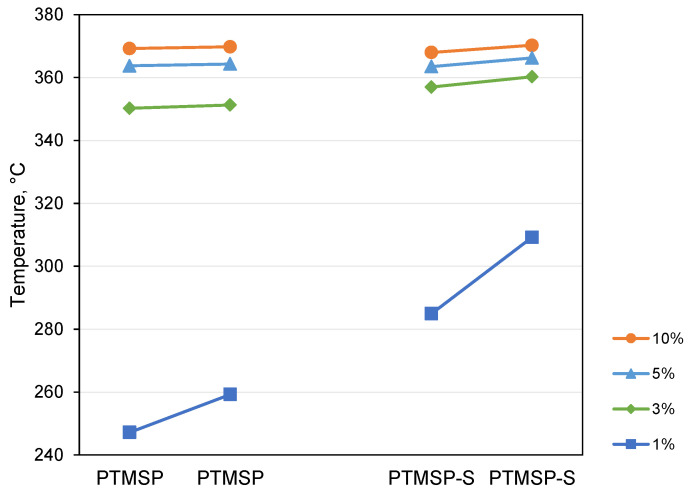
Temperatures of 1, 3, 5, and 10% weight loss of materials under study according to TGA investigation.

**Figure 3 polymers-14-00462-f003:**
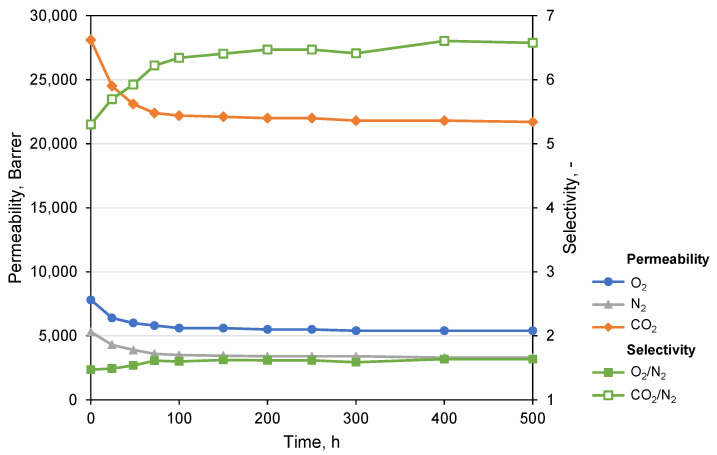
Dependance of the permeability coefficients and ideal selectivity in the PTMSP-S film on total time of heat treatment (140 °C) in air.

**Figure 4 polymers-14-00462-f004:**
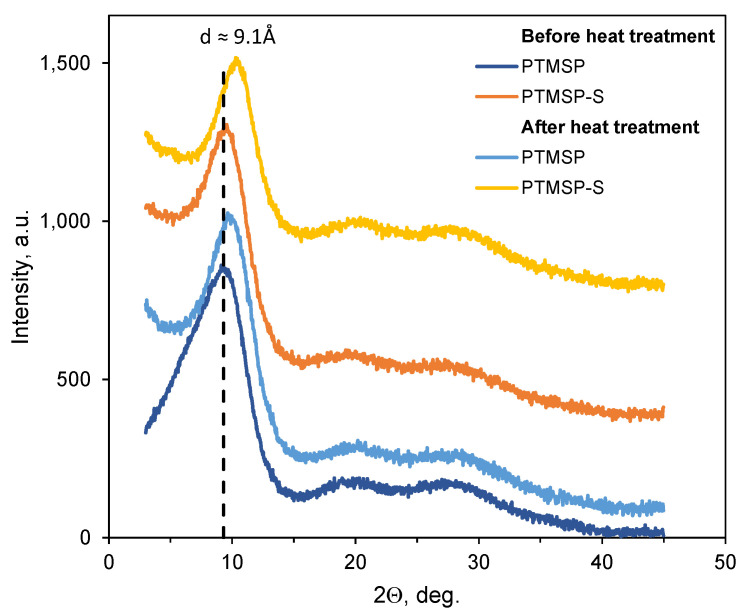
WAXD patterns of pure PTMSP and with stabilizer PTMSP-S before and after heat treatment.

**Figure 5 polymers-14-00462-f005:**
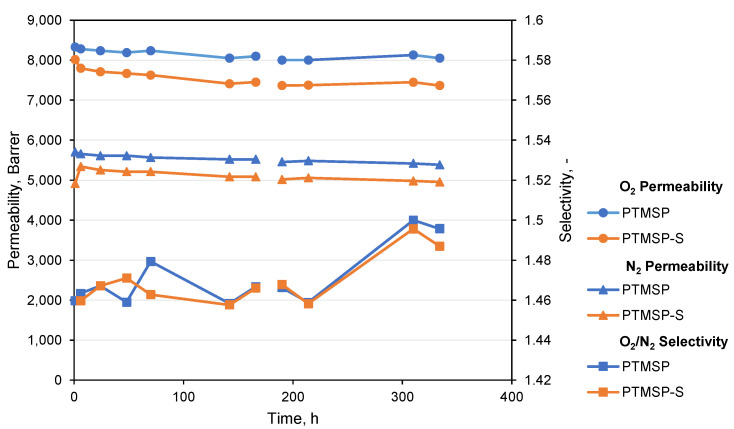
Aging of polymer films at room temperature investigated by O_2_ and N_2_ permeation.

**Figure 6 polymers-14-00462-f006:**
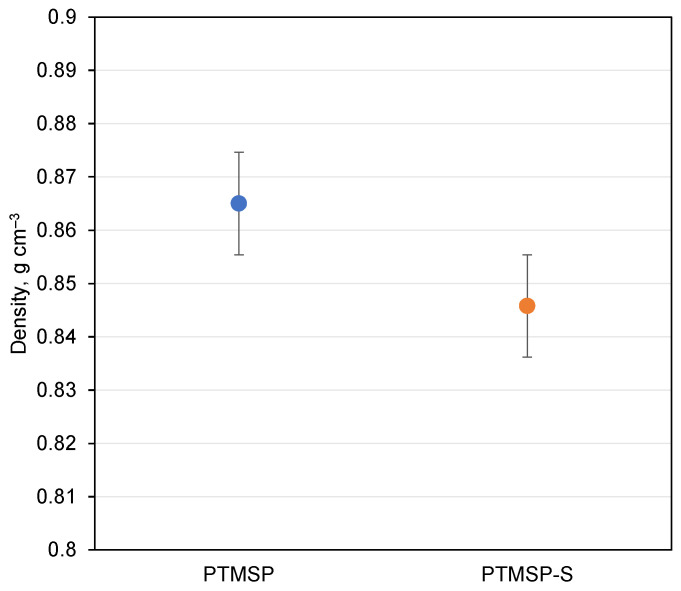
Density of PTMSP films determined from sample geometry.

**Figure 7 polymers-14-00462-f007:**
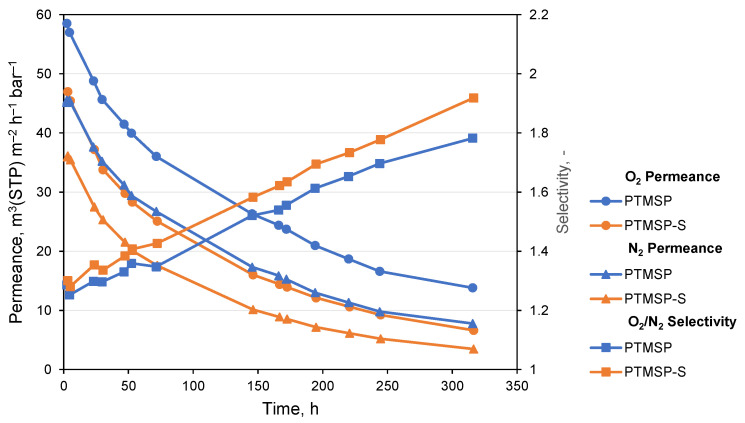
Aging of TFCMs prepared from pure PTMSP and PTMSP-S during an experiment with gas, continuously flowing through membranes.

**Table 1 polymers-14-00462-t001:** Permeability coefficients (P) and ideal selectivity (α) with respect to individual gases for freshly prepared isotropic films of pure PTMSP and PTMSP-S.

Sample	Permeability Coefficient, P (Barrer) ^1^			Ideal Selectivity, α	
	O_2_	N_2_	CO_2_	O_2_/N_2_	CO_2_/N_2_
PTMSP ^2^	8400	5600	30,200	1.5	5.4
PTMSP ^3^	8300	5800	-	2.4	-
PTMSP-S ^2^	7800	5300	28,100	1.5	5.3
PTMSP-S ^3^	8000	4900	-	1.6	-

^1^ 1 Barrer = 1 × 10^−10^ [cm^3^ (STP) cm cm^−2^ s^−1^ cmHg^−1^]; ^2^ Gas permeability was measured for isotropic films prepared by method 1; ^3^ Gas permeability was measured for isotropic films prepared by method 2.

**Table 2 polymers-14-00462-t002:** Permeability coefficients (P) and ideal selectivity (α) with respect to individual gases for films from pure PTMSP and PTMSP-S with added stabilizer Irganox 1076 heat treated at 100 °C.

Sample	Total Time of Heat Treatment, h	Permeability Coefficient, P (Barrer)			Ideal Selectivity, α	
		O_2_	N_2_	CO_2_	O_2_/N_2_	CO_2_/N_2_
PTMSP	24	7100	3700	22,000	1.9	5.9
	48	10,500	8000	27,200	1.3	3.4
	72 ^1^	-	-	-	-	-
PTMSP-S	24	6400	4300	24,500	1.5	5.7
	48	6000	3900	23,100	1.5	5.9
	72	5800	3600	22,400	1.6	6.2

^1^ The sample has become fragile.

**Table 3 polymers-14-00462-t003:** WAXD data of PTMSP films.

Sample	2θ, °	Δ1/2, °	Interchain Distance d, Å
PTMSP (before heat treatment)	9.6	3.69	9.26
PTMSP (after heat treatment)	9.7	3.11	8.87
PTMSP-S (before heat treatment)	9.7	3.56	9.13
PTMSP-S (after heat treatment)	10.5	3.24	8.44

## Data Availability

The data presented in this study are available on request from the corresponding author.

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
