# Peer review of "Effect of Immobilization of Phenolic Antioxidant on Thermo-Oxidative Stability and Aging of Poly(1-trimethylsilyl-1-propyne) in View of Membrane Application"

_polymers, 2022, doi:10.3390/polym14030462_

Round 1

Reviewer 1 Report

Dear authors,

Please find the enclosed file.

Author Response

Response to comments of Reviewer 1:

Thank you for your careful reading of our work and review of our paper. We have answered each of your points below.

Comment 1:

  1. [a. Line 34-35: Due to the nonequilibrium state of PTMSP, physical aging is observed, which a char- acteristic property of glassy polymers is….The sentences needs correction]

Response: We made the corresponding corrections (page 1, lines 35-36).

  1. [Line 66: Nagai and Nak- 66 agawa investigated? Author should provide reference number here. It is better to use technical terms in scientific writing such as “quantity of PC”.]

Response: We made the corresponding corrections (page 2, line 70).

  1. [Line 85: The aim this work is … It needs grammatical correction” The aim of this work is…]

Response: The last part of the Introduction was rephrased, grammar checked. (page 2, lines 77-97).

Comment 2.

Experimental:

  1. [Line 85: Authors should share the synthetic route of the synthesized material.]

Response: The detailed method of polymer synthesis is described in the reference [25]. Synthesis as well as post-synthesis purification and drying of PTMSP samples used in our present article was performed according to the method given in [25]. In case the Reviewer would insist on the polymer synthesis information in the text of the manuscripts it can be added. (Page3, lines 113-117)

  1. 2. [In the section of characterization, authors have overlapped the synthetic process and characterization techniques e.g. after section 2.3.4, authors have provided 2.3.5. Thin film composite membrane (TFCM) formation, It is suggested that authors should first provide all types of thin films or composites formation details then characterization.]

Response: Sections describing experimental methods were thoroughly renumbered for more logic in the manuscript structure.

Results and discussions

[Line 245: Results should be Results and Discussion]

Response: We made the corresponding corrections (page 6, line 258).

  1. [Some of the techniques such as NMR and intrinsic viscosities have been mentioned in characterization but no data has been provided in the Results and Discussion.]

Response: We put at the beginning of the section Results and Discussions the values of intrinsic viscosity and cis-/trans- composition for the samples used in our article and determined by these methods (page 6, lines 260-263).

  1. [Figure 1, The axis on Y axis should be weight loss (%)].

Response: The Figure 1 was changed.

Reviewer 2 Report

Comments to the author:

In this manuscript, the authors presented the effect of immobilization of phenolic antioxidant on thermo-oxidative stability and aging of poly(1-trimethylsilyl-1-propyne). The authors conducted several fundamental analysis of PTMSP and PTMSP mixed with Irganox 1076, however, the contents of experiment are not rigorous, innovative and novel enough to contribute to other researchers. The arguments of this manuscript is not solid and logical. The writings and scientific expressions are not professional and careful. Too many obvious mistake. Overall, I would not recommend the publication of this manuscript.

Major issues:

  1. According to the author’s claim, ‘’the aim this work is to study thermally activated relaxation of isotropic films prepared from PTMSP with addition of antioxidant Irganox 1076 in comparison with pure PTMSP films’’. However, thermally activated relaxation is only inferenced and associated by “Simultaneously with a decrease in permeability, an increase in the ideal selectivity was observed in line 301”. Which is not rigorous and convincing enough. The author should provide more analysis to support this statement. How and why you made this important statement.
  2. In 3.3. Gas transport and structural changes in heat treated PTMSP films, the intensity increase after heat treatment in WARD is not well explained in term of structural changes.
  3. The interplanar distance change for both heat treated pure PTMSP as well as stabilized PTMSP is described in line 322 by Table 3, however, the data provided in Table 3 is Interchain distance. The author should explain the relationship between interplanar distance and Interchain distance for those who are not in your field.
  4. In line 379 to line 382, author should elaborate carefully why t3 elements are too small and t4 elements are sufficient. More experiments or computation process should be conducted or provide to support this statements.
  5. In line 398 to line 401, the thickness of the selective layer can be estimated as 420±50 nm. Please provide the evidence or experiment results to support this estimation. Such as the cross-section view of selective layer by SEM picture. Or the author should use ‘’the thickness of the selective layer is 420±50 nm’’ instead.
  6. The introduction part should include research background, the status of development, and significance of the study and problem statement which are on well provide in this manuscript. The author should rewrite introduction.
  7. In line 112, the characterization condition of intrinsic viscosity should be provided completely. Such as pressure and preparation method which also affect your results. It’s too floppy in current expression.

Minor issues:

  1. Molecular should be expressed in right form:

O2 and N2 should be O2 and N2 in fig.5 and fig.7.

  1. PALS spectra in line 375 should be Positron Annihilation Lifetime Spectroscopy at (PALS) at first time.
  2. Wide-angle X-ray diffraction appears in line 130, 314 and 327. However, WAXD is used in line 329. Please use Wide-angle X-ray diffraction (WAXD) at first time and use abbreviation in following.
  3. In line 33, BET should be Brunauer-Emmett-Teller (BET) at first time.
  4. in line 85, “The aim this work” should be “The aim of this work”.
  5. Reference format should be checked carefully. Year should be bold. Journal abbreviation should be Italic…Please follow the template.

Author Response

Response to Reviewer 2:

Thank you for your perusal of our work and your comments. Our answers to your points are as follows.

Major issues:

  1. [According to the author’s claim, ‘’the aim this work is to study thermally activated relaxation of isotropic films prepared from PTMSP with addition of antioxidant Irganox 1076 in comparison with pure PTMSP films’’. However, thermally activated relaxation is only inferenced and associated by “Simultaneously with a decrease in permeability, an increase in the ideal selectivity was observed in line 301”. Which is not rigorous and convincing enough. The author should provide more analysis to support this statement. How and why you made this important statement.]

Response: We agree that the study of the relaxation of polymer films includes, in addition to changes in gas permeability, a wider range of experimental methods. Therefore, within the framework of our article, it will be more correct to use the term ‘thermal annealing’ in relation to the process of films treatment. We corrected accordingly Abstract, Introduction, and Conclusion sections. At the same time, we have left the term ‘thermally activated relaxation’ where it is related to the reason for the change in packing and gas permeability due to thermal annealing, e.g. in Abstract (page 1, line 20), Introduction (page 2, lines 51-52), Results and Discussions (page 8, line 317).

  1. [In 3.3. Gas transport and structural changes in heat treated PTMSP films, the intensity increase after heat treatment in WARD is not well explained in term of structural changes.]

Response: There is no significant change in the intensity of the main diffraction maximum with an angular position of about 10 degrees (on the 2θ scale) in the diffraction patterns obtained for the films before and after heating. Small changes in the integral intensity of this peak can be associated with different thicknesses of the studied films and, therefore, with different volumes of the substance irradiated by the X-ray beam during the recording of the diffractogram.

  1. [The interplanar distance change for both heat treated pure PTMSP as well as stabilized PTMSP is described in line 322 by Table 3, however, the data provided in Table 3 is Interchain distance. The author should explain the relationship between interplanar distance and Interchain distance for those who are not in your field.]

Response:  This is no doubt a correct remark, since for polymers with an amorphous organization, it is more correct to use the term “interchain distance”. In accordance with this, in the article we have corrected the “interplanar” for “interchain” throughout the manuscript.

  1. [In line 379 to line 382, author should elaborate carefully why t3 elements are too small and t4 elements are sufficient. More experiments or computation process should be conducted or provide to support this statements.]

Response:

Thank you very much for this comment! The role of free volume elements of different sizes and of “bottlenecks” connecting FVEs is of a great interest and importance for understanding of small molecule transport in glassy high free volume polymers! The position of Irganox 1076 molecules in the matrix of PTMSp is of a big question. In order to identify where molecules of this nature and size can be accommodated within the polymer when solvent is gradually removed from the polymer solution it is necessary to carry out separate work using molecules similar to the Irganox 1076 but which can be studied by e.g. photochromic, spin or electrochromic probe techniques. For the current work, taking into account geometry of the Irganox 1076 molecule and of t3 and t4 elements of the free volume (please see the corresponding corrected text part on pages 10 and 11), and taking into account very small amount of antioxidant introduced into the polymer, it was decided to limit consideration of antioxidant placement to t4 free volume elements.

  1. [In line 398 to line 401, the thickness of the selective layer can be estimated as 420±50 nm. Please provide the evidence or experiment results to support this estimation. Such as the cross-section view of selective layer by SEM picture. Or the author should use ‘’the thickness of the selective layer is 420±50 nm’’ instead.]

Response:

Due to the current pandemic situation and thus lack of personal present in the lab, it was decided that electron scanning microscopy of the TFCM cross-section will not be carried out. For the membrane with relatively thick selective layer, it is possible to make estimation of the selective layer thickness based on comparison of permeability coefficient of polymer and membrane permeance. The obtained thickness value of 420±50 nm is significantly bigger than thicknesses of practical optimized TFCMs. For example, the thickness of the selective layer of PDMS TFCM used in Hereon as gutter layer for formation of multilayer membranes is in the range of 150nm, the thickness of PolyActive layer used for CO2/N2 separation is 70-100nm. At the thickness of 420nm one cannot expect strong deviation of polymer properties from those characteristic for thick isotropic films. That is why the thickness was calculated as it is mentioned in the text of the manuscript and this thickness is now marked as “apparent thickness” on page 12, line 417. Of course, one can argue, that calculation of thickness from initial, taken before aging, values of permeability coefficients and permeances is not completely correct, since it would be better to take values at comparable ideal selectivities of films and membranes, but this assumption was considered as sufficiently correct, and we hope the Rviewer will accept it. In out further work all necessary SEM images will be obtained and presented.

  1. [The introduction part should include research background, the status of development, and significance of the study and problem statement which are on well provide in this manuscript. The author should rewrite introduction.]

Response: The last three paragraphs of the Introduction were corrected.

  1. [In line 112, the characterization condition of intrinsic viscosity should be provided completely. Such as pressure and preparation method which also affect your results. It’s too floppy in current expression.]

Response: The corresponding corrections can be found in the section 2.2.2 (page 3, lines 118-122).

Minor issues:

  1. [Molecular should be expressed in right form: O2 and N2 should be O2 and N2 in fig.5 and fig.7]

Response: All figures were corrected.

  1. [PALS spectra in line 375 should be Positron Annihilation Lifetime Spectroscopy at (PALS) at first time]

Response: We made the corresponding correction (page 11, line 391).

  1. [Wide-angle X-ray diffraction appears in line 130, 314 and 327. However, WAXD is used in line 329. Please use Wide-angle X-ray diffraction (WAXD) at first time and use abbreviation in following.]

Response: We made the corresponding corrections, WAXD abreviation is introduced in section 2.3.6. name (page 5, line 199).

  1. [In line 33, BET should be Brunauer-Emmett-Teller (BET) at first time.]

Response: We changed abbreviation BET to Brunauer-Emmett-Teller (page 1, line 33).

  1. [in line 85, “The aim this work” should be “The aim of this work”.]

Response: The last part of the Introduction was rephrased, grammar checked. (page 2, lines 77-97).

  1. [Reference format should be checked carefully. Year should be bold. Journal abbreviation should be Italic…Please follow the template.]

Response: This issue was corrected.

Reviewer 3 Report

The paper presents research on the effect of immobilization of phenolic antioxidant on thermo-ox-idative stability and aging of poly(1-trimethylsilyl-1-propyne). The presentation of methods and scientific results in the current form is satisfactory for publication in the Polymers journal. The minor and significant drawbacks to be addressed can be specified as follows:
1.    Individual sections should be renumbered. I don't know if "2.3.4. Preparation of PTMSP isotropic films" and "2.3.5. Thin film composite membrane (TFCM) formation" (also 2.3.6) should be in "2.3. Characterization". After all, it's the preparation of materials !!! Maybe 2.2.1 and 2.2.2. In general, going into such a large number of subsections seems to be great difficult in reading.
2.    Lines 212. Where did this equation (and Eqs. 2 and 3) come from? Please refer to the literature. 760 mmHg??? What about 76?
3.    Line 250. PTMSP-S with added stabilizer Irganox 1076 ---> PTMSP with added stabilizer Irganox 1076 (PTMSP-S). Nowhere before Tab. 1 does an explanation of the name PTMSP-s appear.
4.    Tab. 2. Please compare the indications in Tables 1 and 2. (i) Permeability, Barrer ---> Permeability coefficient, P (Barrer) (ii) Ideal selectivity a, - ---> Ideal selectivity, α.
5.    Fig. 3. Is this data for sample PTMSP-S? Would you please correct the figure captions?
6.    Fig. 4, Legend. It is not necessary to enter "a" and "b" for PTMSP and/or PTMSP-S. The authors do not use them in the text (see, for example, Tab. 3).
7.    TFCM abbreviation is explained several times in the text. It is sufficient to do only the first time (maybe beyond the abstract). See lines 13, 89, 162, and 388,
8.    I am missing a table that would compare the results (best and/or representative one) taken from Tabs. 1 and 2 with the values obtained for other materials (taken from the literature). Of course, the goal is not to select worse results from the literature but to show how interesting this system is. It would be an excellent summary of the work being reviewed.

Author Response

Response to Reviewer 3:

Thank you for your thoughtful work and your comments. We have answered each of your comments below.

  1. [Individual sections should be renumbered. I don't know if "2.3.4. Preparation of PTMSP isotropic films" and "2.3.5. Thin film composite membrane (TFCM) formation" (also 2.3.6) should be in "2.3. Characterization". After all, it's the preparation of materials !!! Maybe 2.2.1 and 2.2.2. In general, going into such a large number of subsections seems to be great difficult in reading.]

Response: We agree that such numbering and partition of sections complicates the perception of the text. We made another division in the Experimental part that is more appropriate as we think.

  1. [Lines 212. Where did this equation (and Eqs. 2 and 3) come from? Please refer to the literature. 760 mmHg??? What about 76?]

Response: Equations 2 and 3 represent method of permeability coefficient and membrane permeance calculation using data obtained on the “contant pressure / variable volume facility” equipped with the flow sensor Bios DryCal Definer 220 (MesaLabs Inc., Lakewood, CO, USA) This device gives the value of the gas flow in cm3(STP)/min and doesn’t depend on gas nature. The aforementioned equation are derived from simple consideration : how to obtain permeability coefficient (cm3(STP) cm cm-2 s-1 cmHg-1) and permeance (m3(STP) m-2 h-1 bar-1) when one knows gas flow (cm3(STP) min-1 or m3(STP) h-1) through the sample of known area (cm2 or m2), at known pressure gradient (cmHg or bar). In our opinion, this consideration does not need to be referenced to the literature.

Pressure drop across the membrane (P) is measured in atm, then the multiplier 76 is used to convert the value to cm Hg. ‘760’ is the coefficient to obtain the amount of gas (in dimension of cm3(STP)) permeating into the calibrated volume V when pressure in the volume changes for Δp measured in torr. The method of permeability coefficient calculation is taken from the manual for this custom made experimental facility.

  1. [Line 250. PTMSP-S with added stabilizer Irganox 1076 ---> PTMSP with added stabilizer Irganox 1076 (PTMSP-S). Nowhere before Tab. 1 does an explanation of the name PTMSP-s appear.]

Response: We made the corresponding corrections, the term “PTMSP-S” is introduced on page 6, line 264 and used further through the text. Mentioning of Irganox 1076 is left in the text only where it is appropriate.

  1. [Tab. 2. Please compare the indications in Tables 1 and 2. (i) Permeability, Barrer ---> Permeability coefficient, P (Barrer) (ii) Ideal selectivity a, - ---> Ideal selectivity, α.]

Response: Corresponding corrections in Tables 1 and 2 are done.

  1. [Fig. 3. Is this data for sample PTMSP-S? Would you please correct the figure captions?]

Response: Figure caption is corrected.

  1. [Fig. 4, Legend. It is not necessary to enter "a" and "b" for PTMSP and/or PTMSP-S. The authors do not use them in the text (see, for example, Tab. 3).]

Response: Thank you, the Figure is corrected.

  1. [TFCM abbreviation is explained several times in the text. It is sufficient to do only the first time (maybe beyond the abstract). See lines 13, 89, 162, and 388,]

Response: The term “TFCM” is introduced in the abstract (page 1, line 14) and in the main manuscript in the Introduction section (page2 line 96). All excessive mentioning of “thin film composite membrane” was deleted.

  1. [I am missing a table that would compare the results (best and/or representative one) taken from Tabs. 1 and 2 with the values obtained for other materials (taken from the literature). Of course, the goal is not to select worse results from the literature but to show how interesting this system is. It would be an excellent summary of the work being reviewed.]

Response: The properties of PTMSP, including geometric structure (cis-/trans- ratio) and gas permeability, largely depend on the conditions of polymer synthesis (to the greatest extent on the catalytic system). In this work, we used PTMSP obtained on the NbBr5-based catalytic system. We continue experiments to study the aging of PTMSP obtained both on Nb bromide-based systems and on the more traditional and most widely used Nb chlorides. We think, it would be premature to compare the data on PTMSP obtained on other systems under other conditions of aging at this stage of research.

Round 2

Reviewer 1 Report

Please find the enclosed file.

Author Response

Reviewer 1

Dear reviewer,

Thank you very much for your comments! Please find our answers below.

Dear author,

I have positively reviewed the revised version of your article, I appreciate your efforts for the revision, I also acknowledge your positive attitude towards revision, In order to improve the article following points should be considered.

Comment 1. In the response of the major correction regarding synthesis, the reference authors have provided as their own work was published in 2003,  It is better to use some recent reference along with your work to make a good impression on reader.  The authors should also include the synthetic route (Scheme of synthesis).

Answer: Thank you for the suggestion. The synthesis of the PTMSP, carried out according to the procedure described in the reference [25], is considered by our group as completely established method, which does not need any adjustments. Because of this consideration, there isn’t more recent publication containing updated method description.

Comment 2.  In the response of the major correction regarding NMR and Intrinsic viscosity data, author should provide the spectra.

Answer: The 13C-NMR  spectra of PTMSP is included into the Supporting Information file. The reference to the Figure S1 is included into the main text Page 3 line 142. No additional information on intrinsic viscosity determination can be added neither into the main text nor into the Supporting Information.

Reviewer 2 Report

No more question.

Author Response

Reviewer 2

Comments and Suggestions for Authors

No more question.

Answer: Dear reviewer, thank you for your positive consideration of the work done in order to follow your previous comments!

Reviewer 3 Report

The authors have addressed my queries satisfactorily. The manuscript was improved in accordance with my suggestions and I have no further objection to this paper. The quality of the paper has been improved and, therefore, I consider that the article can be accepted in its present form.

Author Response

Reviewer 3

Comments and Suggestions for Authors

The authors have addressed my queries satisfactorily. The manuscript was improved in accordance with my suggestions and I have no further objection to this paper. The quality of the paper has been improved and, therefore, I consider that the article can be accepted in its present form.

Answer: Dear Reviewer, thank you for this very positive comment!